# Resolving out of Africa event for Papua New Guinean population using neural network

Mayukh Mondal [1,2] ✉, Mathilde André [2,3,4], Ajai K. Pathak [3,5], Nicolas Brucato[6], François-Xavier Ricaut [6], Mait Metspalu [3] & Anders Eriksson [2]

The demographic history of the Papua New Guinean (PNG) population is a subject of interest due to its early settlement in New Guinea, its relative isolation and substantial Denisovan ancestry. Previous research suggested an admixture with an early diverged out of African population. This study re-examines the PNG population using newly published samples. Our findings demonstrate that the observed shifts in Relative Cross Coalescent Rate (RCCR) curves are driven by strong bottleneck and slower population growth rate of the PNG population, rather than the contributions from an earlier out of Africa population. Although a small contribution from the early out of Africa population cannot be ruled out, it is no longer needed to explain the observed results. Our analysis positions them as a sister group to other East Asian populations. This study provides insights on the PNG population and highlights the impact of population-specific demography on interpreting RCCR curves.

The Papua New Guinean (PNG) population is among the most fascinating in the world, owing to its unique demographic history. Following the Out Of Africa (OOA) event, modern humans populated New Guinea at a remarkably early date-at least 50 thousand years ago[1]. Since then, the population has remained relatively isolated compared to other OOA populations (such as European and East Asian populations)[2–5] and has gone through a strong bottleneck[6]. The substantial Denisovan ancestry within the PNG population[7,8] and the strong correlation between Denisovan and Papuan ancestries[9], contribute to the genetic distinctiveness of the PNG population.

Researchers have suggested that the genomes of PNG populations contain evidence of admixture with a modern human population that might have diverged from African populations - around 120 thousand years ago—much earlier than the proclaimed primary divergence between African and OOA populations[4,10]. However, the extent to which this early diverged population contributed to the genome of PNG populations remains a subject of ongoing debate[3,4,11,12].

Interestingly, this early migration hypothesis is more widely accepted by archeologists[13–17].

Pagani et al.[4] supports this hypothesis, notably through Relative Cross-Coalescent Rate (RCCR) analysis. This RCCR analysis suggests that the PNG population diverged from African populations significantly earlier than other OOA populations. They argued that this earlier divergence indicated by the RCCR curve might reflect a contribution from an earlier OOA population specific to PNG. While this shift in the RCCR curve is well-documented[2,3,5], some researchers attribute it to technical artefacts such as low sample sizes and phasing errors rather than genuine demographic events.

The origins of the primary lineage of the PNG population have also been contested. Some researchers propose that the PNG population is closely related to the Asia-Pacific populations and serves as a sister group to other Asian populations[11,12,18]. Conversely, other researchers argue that the PNG population is an outgroup to both European and East Asian populations[3,9,19].

[1]Institute of Clinical Molecular Biology, Christian-Albrechts-Universität zu Kiel, Kiel, Germany. [2]Centre for Genomics, Evolution & Medicine, Institute of Genomics, University of Tartu, Tartu, Tartumaa, Estonia. [3]Estonian Biocentre, Institute of Genomics, University of Tartu, Tartu, Tartumaa, Estonia. [4]Deptartment of Neurosciences "Rita Levi Montalcini", University of, Turin, Italy. [5]Department of Human Genetics, KU Leuven, Leuven, Belgium. [6]Centre de Recherche sur la Biodiversité et l'Environnement (CRBE), Université de Toulouse, CNRS, IRD, Toulouse INP, Université Toulouse 3—Paul Sabatier (UT3), Toulouse, France. ✉e-mail: mondal.mayukh@gmail.com

Recent advancements in analytical methods may provide insights into these debates. For example, Approximate Bayesian Computation with Deep learning and sequential Monte Carlo (ABC-DLS) allows for the use of any summary statistics derived from simulations to train neural networks, which can then predict the most likely demographic models and parameters based on empirical data[20]. Additionally, the Relate software[21] enhances RCCR analysis by employing a modified version of the hidden Markov model, initially used in the Multiple Sequentially Markovian Coalescent (MSMC) method[22,23], allowing for the analysis of thousands of individuals with greater robustness.

In this paper, we re-examine the demographic history of the PNG population using recently published samples[24] combined with data from the 1000 Genome Project[25,26] and cutting edge methods[20,21]. This approach has enabled us to address these longstanding questions with greater precision. We first generate empirical RCCR curves and demonstrate that the previously observed shift is unlikely to be the result of low sample size or phasing errors. Through simulations, we further show that the PNG population is indeed a sister group to East Asian populations, and this shift is probably not due to contributions from an earlier OOA population. Instead, it is likely a consequence of a strong bottleneck and slower population growth in the PNG population.

## Results

### Analysis of empirical data

We used Relate[21] to estimate the effective population size and track RCCR changes over time. To ensure robustness, we performed this analysis using 40 random samples per population, repeated 10 times to generate confidence intervals (Fig. 1). The results for effective population size (Supplementary Fig. 1) align closely with previous findings[2,5,21]. However, RCCR analysis reveals a significant earlier divergence between the PNG highlander population and African population (using Yoruba as the reference) around 100,000 years ago, with a RCCR value of 68% (95% Confidence Intervals = 66–70%). This contrasts with other OOA populations such as Europeans (83% [95% Confidence Interval = 81–85%]) and East Asians (83% [95% Confidence Interval = 82–84%]) at the same time point. A similar shift was also found when lowlanders from PNG were used (Supplementary Fig. 2).

This earlier separation in PNG has been noted by various studies using MSMC analyses[2,4,5,22]. Pagani et al.[4] suggested that this shift might be due to a contribution from the first OOA population to the PNG, while others have speculated that it could be due to phasing errors in the PNG genome[2,27]. Given that we phased the genomes using 249 samples from PNG (Supplementary Data), the likelihood of systematic phasing errors causing this shift is minimal[28] and is therefore an unlikely explanation for this shift as claimed before[2,27]. This conclusion is further supported by analyses using physically mapped datasets (Supplementary Fig. 3) from the SGDP[27], although these results are less definitive due to the smaller sample size ($n = 2$ per population). A similar shift (Supplementary Fig. 4) was also noted in the Andamanese population[12] though the sample size for Andamanese is smaller as well ($n = 9$). The Andamanese are particularly interesting as they do not have sizable Denisovan ancestry or contributions from an earlier OOA population[11,12,18], which helps to narrow down the possible causes for the RCCR shift.

### Reconstructing demographic history of PNG

To explore the demographic processes causing the observed RCCR shift, we tested five plausible demographic scenarios labelled A, O, M, AX, and OX (Fig. 2). In Model A, the PNG and East Asian populations are sister groups[11,12,18]. Model O positions the PNG population as an outgroup to both European and East Asian populations[3,19]. Model M combines elements of both A and O, proposing that the PNG population arose from admixture between a sister group of the East Asian population and an outgroup of European and East Asian populations. Model AX postulates that the PNG population is a sister group to the

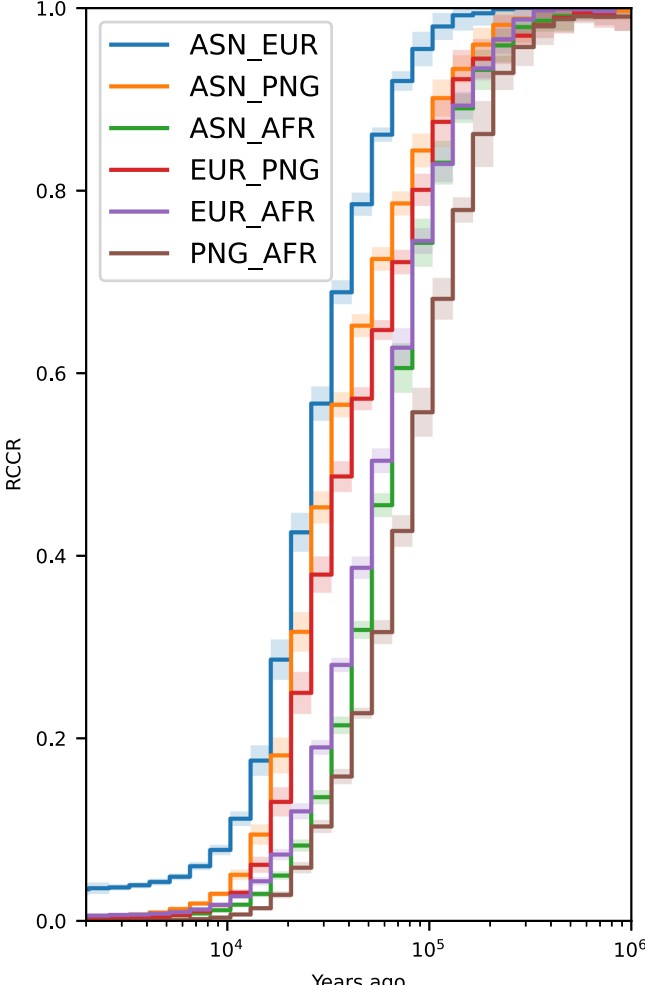

**Fig. 1 | Relate results of relative cross coalescent rate (RCCR) curve on empirical data using 40 samples per population.** AFR = Yoruba, EUR = British from England and Scotland, ASN = Han Chinese, and PNG = Highlanders of Papua New Guinea. The x-axis is years ago from modern times with the log scale. The y-axis is RCCR values calculated by Relate. The shaded regions are the 95 percent (mean ± two standard deviation) confidence intervals created from 10 separate random sampling events. The line represents the mean values.

East Asian population but received input from an earlier OOA population[4]. Finally, in Model OX, the PNG population received a contribution from an earlier OOA population, while remaining ancestry came from an out group to the European and East Asian populations (see methods for further details).

We trained our neural network using cross-population Site Frequency Spectrum (cSFS) data under these five simulated models (Supplementary Table 1, Fig. 2). Empirical data indicated that Model A to be the most accurate representation of the demographic history with a high $p$ value > 0.05 for goodness of fit analysis (Supplementary Table 2). This result held even after excluding genic regions with 10 kb flanking areas, suggesting that the selective pressure working on the genome had minimal impact for our model decision (Supplementary Table 3). However, it is important to note that a low contribution (less than 5%) from an early OOA or outgroup of Eurasia could be misinterpreted as Model A (Supplementary Figs. 5, 6) but Model OX can be rejected with high certainty regardless of the admixture amount (Supplementary Fig. 7).

We also ran models with low migration rates ($m < 5 \times 10^{-5}$, where m is the proportion of individuals moving from one population to another per generation) between modern human populations after

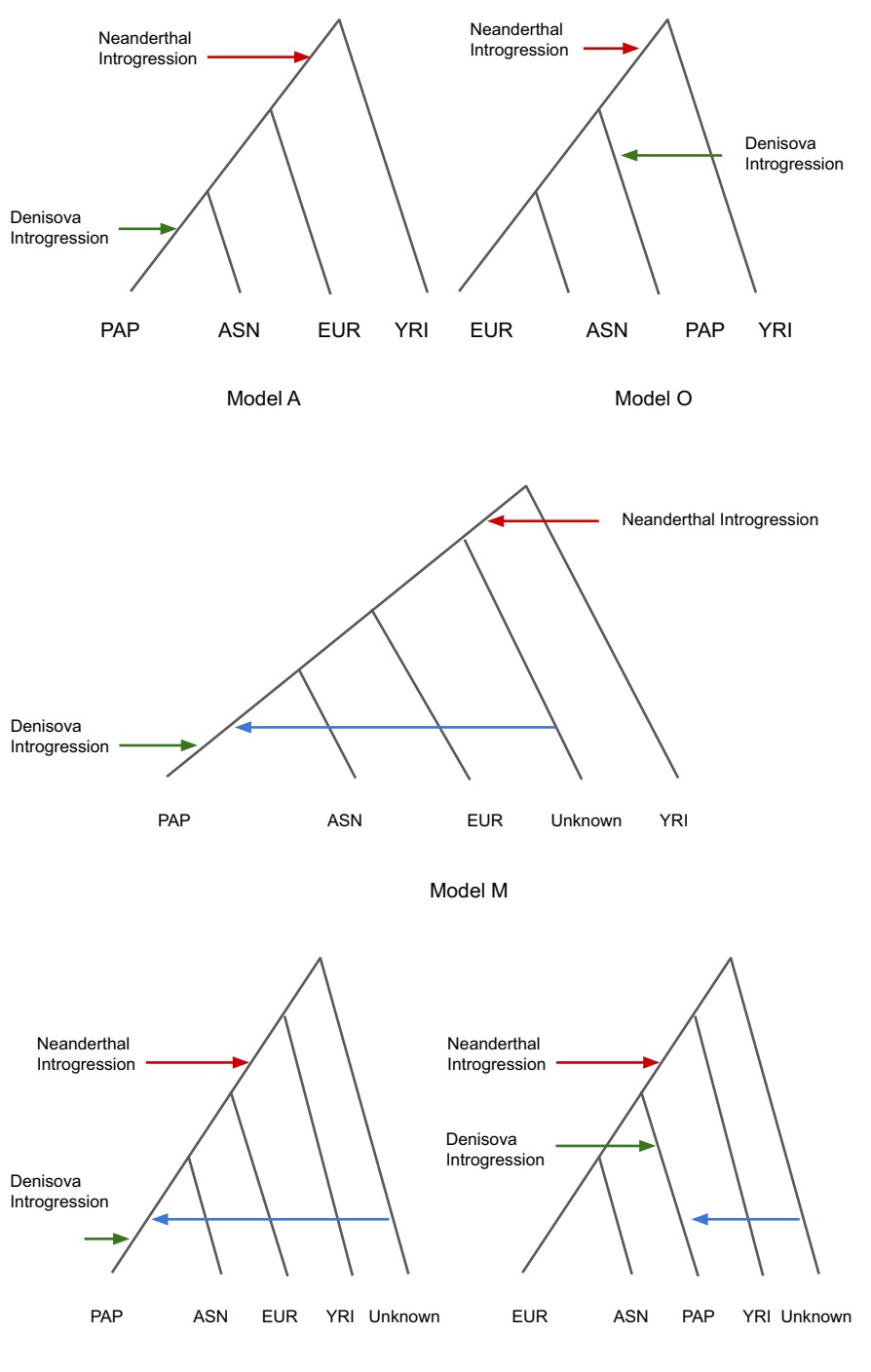

**Fig. 2 | Simplified schematic of demographic models tested in this Study.** AFR Africa, EUR European, ASN East Asian, PAP Papua New Guinean, Model A Papua New Guineans and East Asians are sister groups, Model O Papua New Guineans are an outgroup of Europeans and East Asians, Model M a mixture between model A and O, Model AX Early out of Africans contributed to Papua New Guineans who are a sister group to East Asians. Model OX Early out of Africans contributed to Papua New Guineans who are an outgroup of Europeans and East Asians. Red arrow represents the Neanderthal introgression for all out of Africa populations, and the green represents the Denisova introgression for Papua New Guinean.

they were separated from each other, and the results remained consistent (Supplementary Table 4). However, when we increased the migration rate ($m < 5 \times 10^{-4}$), although Model A remained the top-ranked model, the evidence for Model O was not low (Bayes factor between Model A and O [$BF_{AO}$] < 10, Supplementary Table 5). The higher migration rates lead to equifinality in the cSFS, which the neural network failed to distinguish[20,29] and generally gave inconsistent results when running multiple times.

The best-fitting parameters for Model A (Table 1, Fig. 3) largely correspond with the previously established OOA model, with some deviations specific to the inclusion of the PNG population[20,30–32]. Our model suggests that all OOA populations, including PNG, diverged from African populations (represented by Yoruba) around 62.4 (95% Credible Intervals = 62–62.8) thousand years ago, experiencing a strong bottleneck (Table 1). Approximately 52 (95% Credible Intervals = 51.6–52.8) thousand years ago, Neanderthals contributed

**Table 1 | Best fitted parameter values of model A**

| Params | Description | Mean (CI) | Events (kya) |
|---|---|---|---|
| $N_A$ | The ancestral effective population size | 19,308 (19,285–19,318) | |
| $N_{AF}$ | Effective population size of modern African population | 43,477 (43,048–43,957) | |
| $N_{EU}$ | Effective population size of modern European population | 71,931 (68,012–74,750) | |
| $N_{AS}$ | Effective population size of modern East Asian population | 94,703 (89,594–98,160) | |
| $N_{PA}$ | Effective population size of modern PNG population | 32,085 (30,889–33,303) | |
| $N_{NE}$ | Effective population size of Neanderthals | 2360 (2347–2378) | |
| $N_{DE}$ | Effective population size of Denisova | 2535 (2519–2549) | |
| $N_{EUO}$ | Effective population size of European population before exponential growth | 3512 (3423–3589) | |
| $N_{ASO}$ | Effective population size of East Asian population before exponential growth | 1771 (1730–1799) | |
| $N_{PAO}$ | Effective population size of PNG population before exponential growth | 674 (663–689) | |
| $N_B$ | Effective population size of OOA population | 726 (718–735) | |
| $N_{ND}$ | Effective population size of archaic hominin population | 15,324 (15,183–15,393) | |
| $T_{DPM}$ (ky) | Time interval of introgression of PNG population from Denisova | 31.3 (31.1–31.5) | 31.3 (31.1–31.5) |
| $T_{AS\_PA}$ (ky) | Time interval of separation between East Asian and PNG | 14.9 (14.6–15.1) | 46.2 (45.9–46.5) |
| $T_{EU\_AS}$ (ky) | Time interval of separation between East Asian and European | 5 (4.7–5.1) | 51.2 (50.8–51.6) |
| $T_{NOM}$ (ky) | Time interval of introgression of OOA population from Neanderthal | 0.9 (0.8–1) | 52 (51.6–52.8) |
| $T_B$ (ky) | Time interval of separation of African and OOA population | 10.3 (10.2–10.4) | 62.4 (62–62.8) |
| $T_{AF}$ (ky) | Time interval of changes in effective population size of humans from ancestral population size | 43.2 (41.8–44.6) | 105.6 (104.1–107) |
| $T_{NI\_NS}$ (ky) | Time interval of separation between sequenced Neanderthal and introgressed Neanderthal | 200.6 (198.6–203.3) | 252.6 (250.2–255.1) |
| $T_{DI\_DS}$ (ky) | Time interval of separation between sequenced Denisoval and introgressed Denisova | 276 (274.6–277) | 307.4 (306.2–308.6) |
| $T_{N\_D}$ (ky) | Time interval of separation between Neanderthal and Denisova | 14.3 (12.8–15.5) | 321.6 (319.8–323.4) |
| $T_{H\_A}$ (ky) | Time interval of separation of between modern humans and archaic hominins | 269.8 (268.8–271.4) | 591.5 (589.3–593.7) |
| $^{NEA}m_{OOA}$ (%) | Amount of Neanderthal introgression to OOA population | 4.04 (3.94–4.11) | |
| $^{DEN}m_{PAP}$ (%) | Amount of Denisova introgression to PNG population | 3.23 (3.1–3.34) | |

CI: Credible interval, ky: thousand years and kya: thousand years ago. PNG is Papua New Guinean and OOA is Out Of Africa.

around 4.04% (95% Credible Intervals = 3.94–4.11%) of the genome to these OOA populations. Shortly thereafter, Europeans and East Asians diverged from the PNG populations around 51.2 (95% Credible Intervals = 50.8–51.6) and 46.2 (95% Credible Intervals = 45.9–46.5) thousand years ago, respectively. The PNG population then mixed with Denisovans around 31.2 (95% Credible Intervals = 31.1–31.5) thousand years ago, contributing approximately 3.23% (95% Credible Intervals = 3.1–3.34%) to the genome of PNG. Our analysis also shows that the PNG population experienced a more severe bottleneck (674 [95% Credible Intervals = 663–689] of effective population size) than other OOA populations (i.e. Europeans 3512 [95% Credible Intervals = 3423–3589] and East Asians 1771 [95% Credible Intervals = 1730–1799] of effective population size), with growth rates lower than those of other OOA populations, consistent with previously published data[6,33,34]. While our parameter inference is generally robust within the individual model, substantial changes occur when the underlying model is altered. Given that determining the precise demographic model for human populations is an ongoing effort, parameter estimates should be considered supplementary to the model rather than independent results.

### Demographic model of PNG and RCCR curves

RCCR curves were produced using Relate[21] on simulated data coming from the best-fitting parameter values of Model A (Table 1). These curves replicated the empirical shift towards older divergence times for PNG versus Yoruba compared to other OOA populations (Fig. 4). This shift appears to be primarily driven by the severe bottleneck and lower growth rate experienced by the PNG population (Supplementary Fig. 8). Notably, a simpler model without any archaic introgression (with all other parameters remaining the same as in Table 1 except introgression) produced a similar shift (Supplementary Fig. 9). When we simulated an African population with an earlier divergence, around 300 thousand years ago (similar to the San population)[35], the RCCR

shift became less pronounced as observed before[3,4], indicating that the timing of African population divergence also influences the results (Supplementary Fig. 10).

To further investigate this phenomenon, we conducted simulations of the PNG population under varying bottleneck intensities and growth rates using a minimal model to delimit the effects (see Methods for more details, Supplementary Table 6). The RCCR shift is influenced by both the bottleneck and the growth rate of the PNG population (Supplementary Fig. 11). Further analysis of the most pronounced RCCR shift (produced from a bottleneck of 500 individuals and no exponential growth for PNG) revealed that the estimates of coalescent rates for the PNG population, both within their population and cross population with Africa, are affected by this phenomenon. This in turn impacts the RCCR curve (Supplementary Fig. 12), as RCCR is a ratio of these coalescent rate estimates [$RCCR_{01} = 2 \times \lambda_{01}/(\lambda_0 + \lambda_1)$, where $\lambda$ represents the coalescent rate, and 0 and 1 denote the two different populations]. The shift is primarily caused by changes in the coalescent rate estimates of PNG, which were affected due to the demographic history (changes in effective population size) occurring much before the actual event, extended over a much longer time period than the event itself[36]. Although the bottleneck of the PNG occurred around 50 thousand years ago, this event affected the coalescent rate estimate more than 100 thousand years ago. Thus, the RCCR estimate deviated more for PNG than Europeans.

A similar pattern was observed in the RCCR curves between the PNG and other OOA populations (Fig. 4). Although Europeans are the true outgroup of East Asian and PNG populations in our simulated model, the RCCR analysis shows a greater separation between PNG and Europeans or East Asians than between Europeans and East Asians. This observation in empirical RCCR analysis was a key factor in the initial hypothesis that PNG are an outgroup to Europeans and East Asians[4].

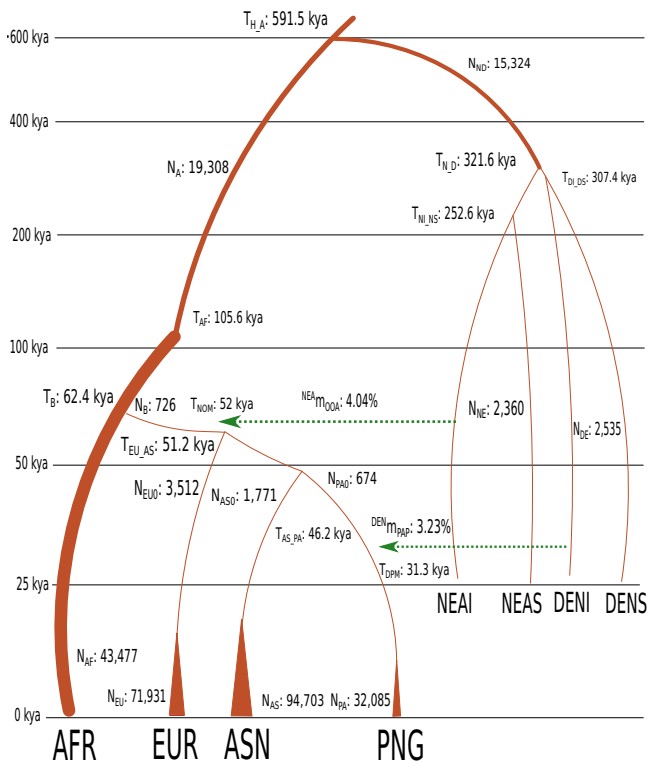

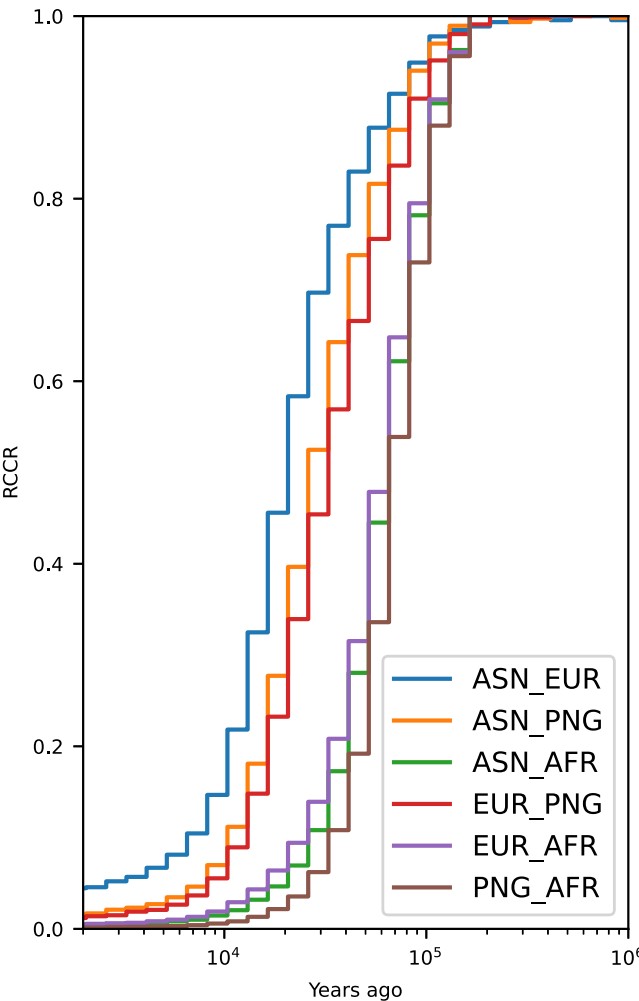

**Fig. 3 | The schema of the Model A with the best fitted mean values of parameters.** AFR Africa, EUR European, ASN East Asian, PAP Papua New Guinean, NEAI introgressed Neanderthal, NEAS sequenced Altai Neanderthal, DENI introgressed Denisova, and DENS sequenced Denisova. The corresponding values are written with the name of the parameters. The graph is not according to the scale. The y-axis represents time events from a thousand years ago (kya) from now.

**Fig. 4 | Relate results of relative cross coalescent curve on simulated data of best fitted values from Model A using 40 samples per population.** AFR African, EUR European, ASN East Asian, and PNG Papua New Guinean. The x-axis is years ago from modern times with the log scale. The y-axis is RCCR values calculated by Relate.

## Discussion

We successfully replicated the shift observed by Pagani et al.,[4] confirming its presence in both physically mapped and statistically phased sequences, which involved over 100 PNG samples. This consistency suggests that the shift is reproducible, though its underlying cause may differ from the original interpretation of Pagani et al.[4] Our analysis using ABC-DLS supports a simpler demographic model for PNG populations, proposing them as a sister group to East Asians with no substantial detectable contribution from an earlier OOA population. This result was reproduced across three independent cSFS datasets, each analyzed 10 times to assess the robustness of our findings. With the exception of the high migration rate model, where differentiation power between models was low, our results remained consistent. The robustness of our findings suggests that increasing the number of cSFS datasets from empirical data is unnecessary for this context.

Model A, with the best-fit parameters (Table 1), aligns well with empirical data in both the Relate results (Supplementary Figs. 13, 14, 15) and the allele frequency spectrum (Supplementary Fig. 16). Since the SFS is not influenced by local recombination rates[37], our model inferences are not biased by patterns of linkage disequilibrium (LD). Therefore, a comparison between empirical and simulated LD patterns was not necessary.

Interestingly, our simulated models reveal that a stronger bottleneck with a lower growth rate could produce a similar shift in RCCR analysis and potentially be misinterpreted as a signal of an earlier population separation. While RCCR is a valuable proxy for estimating the separation time between populations, it is not without biases. The shift could result from various factors, including earlier divergence times[22], admixture with earlier diverged populations[4], or even a

bottleneck in one of the populations, as demonstrated in our study. This demographic history of stronger bottleneck with slower growth rate was also experienced by the Andamanese population, which explains the shift found in the Andamanese population as well[38]. Thus, using RCCR analysis to rebuild the tree of divergence might need to be revised[39,40].

The observed shift in the RCCR curve suggests that a recent bottleneck can impact estimates of effective population size in the distant past. Notably, in our simulations, the Papua New Guinean bottleneck occurred much later (around 46.2 thousand years ago, as shown in Table 1) than the observed shift (peaking around 100 thousand years ago, as depicted in Fig. 4) with a population (Yoruba) that separated a long time ago. This finding implies that the estimation of effective population size and cross-coalescent rates may not be entirely independent, potentially affecting RCCR analysis in its current form, as also observed before[41]. Further analysis suggests (Supplementary Fig. 12) that the estimation of coalescent rate was affected earlier than the true changes of effective population size, which shifts the RCCR curve as RCCR is a ratio of coalescent rates. Additionally, this shift was absent in simulations involving populations that separated 300 thousand years ago, akin to the San population, indicating that the bottleneck effect diminishes over longer separation times (Supplementary Fig. 10).

Our results also reveal that when the contribution from an earlier OOA population is between 1–5%, our neural analysis misclassifies the Model AX to be Model A at a higher rate (Supplementary Fig. 5). A similar issue arises with Model M, where a low contribution (less than 5%) from an outgroup Eurasian population can still be misclassified as Model A (Supplementary Fig. 6). Thus, our analysis does not work for less than 5% contribution from these unknown ghost populations, though Model OX does not show a similar phenomenon with Model A misclassification (Supplementary Fig. 7). While we cannot completely rule out the possibility of a small contribution from these populations, our analysis suggests that such models are not necessary to explain the RCCR shift as previously proposed[4].

Our parameter estimation suggests that the PNG population separated from other populations around 46.2 (95% Credible Intervals = 45.9–46.5) thousand years ago, a timeline that aligns with archaeological estimates of when the ancestors of PNG reached the ancient continent of Sahul, the landmass that once connected New Guinea and Australia[1]. Additionally, our Relate analysis indicates that the separation time between PNG and European populations was the longest observed among OOA populations. However, as our model suggests, this is likely a bias caused by the bottleneck of PNG. This bottleneck may lead to an overestimation of the separation time, particularly in RCCR analysis. In reality, it is more likely that PNG and East Asian populations separated later than the divergence between PNG and European populations.

There are two competing hypotheses about the primary lineage of PNG. One position PNG as a sister group to East Asian populations, while the other places it as an outgroup to both Europeans and East Asians. When a proper model comparison was claimed to be used, a key difference between these hypotheses lies in the modeling approach. Models supporting the sister group hypothesis generally exclude constant migration rates between populations[42,43], while those supporting the outgroup hypothesis include substantial migration rates[3].

Models without migration may be incomplete, but our results consistently position PNG as a sister group to East Asian populations[11,12,18,42,43] even in cases where small migration rates are incorporated ($m < 5 \times 10^{-5}$ individuals per generation). Our Model O closely resembles outgroup-based models[3,9,19]. Given that those models used substantial migration rates, especially for ancestral populations which we lack ($m > 5 \times 10^{-5}$ individuals per generation as mentioned in stdpopsim v0.3.0)[44,45], they are not directly comparable to our models. Indeed, with high migration rates even for modern populations, our approach failed to distinguish between Model A and O with high certainty (Supplementary Table 5).

Nonetheless, our work suggests that the main lineage of PNG is coming from a sister group of Asia, which was not confounded by a convoluted ancestral migration rate pattern between populations. Currently, migration rates among ancestral populations cannot be estimated in a model-free way. This limits the credibility of models using high migration rates between ancestral populations and increases the risk of overfitting if the model structure is not sound.

Given fossil evidence of early human settlement in South and Southeast Asia—predating that of Europe or East Asia[46]—it is tempting to use high migration rates between ancestral populations to better align models with these records, often supporting model O or X. However, it remains uncertain whether the substantial migration rates (i.e., $m > 5 \times 10^{-5}$, as proposed by outgroup models) between ancestral populations over long periods and across large geographic distances were realistic. Current models without migration rates, including those incorporating ancient genomes, do not support model X or O[42,43]. While ancestral populations likely exchanged some gene flow, these migration rates alone should not be the main explanation for the population branching pattern of PNG—especially without a direct, model-free way to confirm it. This hypothesis of high ancestral migration rates should be revisited when ancient genomes from those periods become available by recalculating the migration rates without relying on predefined models.

In conclusion, our study provides compelling evidence that the unique demographic events—specifically a strong bottleneck and slower population growth—within the PNG population are key factors influencing the observed shifts in RCCR curves. These findings not only refine our understanding of PNG's demographic history but also emphasise the necessity of accounting for population-specific demographic events when interpreting RCCR curves.

## Methods

### Ethics

This study was approved by the Research Advisory Committee of University of Tartu and by the Medical Research Advisory Committee of Papua New Guinea under research ethics clearance MRAC 16.21 and the French Ethics Committees (Committees of Protection of Persons CPP 25/21_3, n_SI: 21.01.21.42754). Permission to conduct research in PNG was granted by the National Research Institute (visa n°99902292358) with full support from the School of Humanities and Social Sciences, University of Papua New Guinea.

### Relate

**Data.** This study did not generate any sequence data. For our analysis using Relate, we utilised a previously published Variant Calling Format (VCF) file[24]. This file was filtered and phased following the methods outlined in the original publication. We kept the sites with a minimum base quality of 20, an alignment minimum mapping quality of 20, and downgrading mapping quality for reads containing excessive mismatches with a coefficient of 50. On top of that, we excluded indels, sites with more than two alleles, sites with a maximum missing rate of 5%. We removed sites whose depth of coverage summed across all samples was higher or lower than the sum of the average depth across the dataset by a factor of 2-fold.

We phased the data using SHAPEIT v4.2[47], opting for statistical phasing given the ample sample size. For further analysis, we retained only the Yoruba, British from England and Scotland, Han Chinese South[25,26], both highlander and lowlander populations from PNG[24], and Andamanese population[12] (see Supplementary Table 7 for details).

After phasing, we only kept 40 random samples per population to make it equal (except for analysis of Andamanese, in which case we used 9 samples per population). This random sampling process was repeated 10 times to calculate confidence intervals.

**Relate analysis.** We then used Relate_v1.1.8[21] to generate genealogical trees and estimate the effective population size across all populations. Our analysis revealed that running the analysis on chromosome 1 alone produced results comparable to those obtained from the entire genome. Therefore, we limited our analysis to chromosome 1. To account for the impact of sample size on the results[21], we maintained an equal sample size across populations. For cases with two or 9 samples per population (Supplementary Fig. 4) chromosome 1 might not be enough to get robust results. Thus, we analysed all autosomal chromosomes to increase statistical power.

For the physically mapped data, we downloaded 10x Genomic Variant Calling Format (GVCF) data for two samples per population from the respective source[27]. The individual samples were merged using BCFtools merge[48] and then analysed with Relate across all chromosomes.

### ABC-DLS

Approximate Bayesian Computation with Deep Learning and Sequential Monte Carlo (ABC-DLS) combines ABC with neural networks for dimensionality reduction and uses Sequential Monte Carlo (SMC) for recursive refinement.

ABC is a Bayesian framework that compares summary statistics between simulated and empirical data without relying on a likelihood function. This flexibility allows it to work with any summary statistics. However, ABC suffers from the curse of dimensionality, where performance declines as the number of summary statistics increases. To address this, we used neural networks—a supervised machine learning method—to reduce dimensionality.

First, we trained the neural network using simulated data, with summary statistics as input and either model identity or parameter values as output. Once trained, the network was applied to empirical data to predict the most likely model or parameters. ABC was then used on the neural network output (which can be seen as second-order summary statistics in this case) to calculate model probabilities or parameter credible intervals.

To further narrow the credible intervals, we applied SMC for iterative refinement. Starting with a prior range for the parameters, we generated a posterior range using the neural network and ABC. This posterior range then became the prior range for the next cycle. These SMC cycles were repeated until convergence, defined as the point where the posterior range no longer shrinks significantly compared to the prior range.

**Data.** All position VCF files were required before merging with archaic genomes to produce accurate Site Frequency Spectrum (SFS). Thus, we downloaded GVCF files coming from the high coverage 1000 genome dataset[25,26] for the Yoruba ($n = 15$) to represent African population, British from England and Scotland ($n = 15$) to represent European population, and Han Chinese South ($n = 15$) to represent East Asian population. Additionally, we used fastq files from PNG highlanders ($n = 15$) from our previous publication[24]. The samples were randomly selected from these datasets (see Supplementary Table 7). We also obtained all-position VCF files for Neanderthal from the Altai and Denisovan from their respective repositories[7,49], and ancestral information was downloaded from the 1000 Genomes data[50]. Both Altai Neanderthal and Denisovan are ancient archaic individuals who diverged from modern humans around 600 thousand years ago[51]. They separated from each other around 400 thousand years ago and populated Western and Eastern Eurasia, respectively. Neanderthal is believed to have contributed to all out of Africa populations, whereas Denisova only contributed substantially to PNG[7], Australian Aboriginal[3], and Philippine Ayta[52] populations.

**Mapping.** The fastq files from PNG samples were mapped to the GR38 reference genome using BWA-MEM v0.7.12[53]. The resulting BAM files were sorted and merged (for samples with multiple runs) using SAMtools v1.9[48,54]. Duplicates were marked using Picard tools from GATK v4.2.0.0[55]. and base recalibration was performed using Base-Recalibrator and ApplyBQSR, both from GATK. Finally, the BAM files were converted to CRAM format using SAMtools, adhering closely to GATK's best practices[56].

**Variant calling.** We performed variant calling on the PNG CRAM files using the HaplotypeCaller from GATK to generate GVCF files. Since an unbiased SFS required all-position VCF files, we conducted variant calling using mpileup from BCFtools for all modern samples[57].

**Liftover.** Neanderthal and Denisovan VCF files, originally aligned to GR37, were lifted over to GR38 using LiftoverVcf from GATK[55]. We then merged the all-position VCF files from modern human genomes with those from archaic genomes using BCFtools. The resulting VCF files were used to create an unbiased SFS for both modern and archaic populations. This strategy helped us to retain variant sites that were mono-allelic or fixed in modern humans but not in archaic genomes. We filtered for biallelic SNPs present in every individual and annotated the VCF files with ancestral information using BCFtools query and annotate commands.

**Cross-population site frequency spectrum.** We filtered out positions with insufficient power for variant calling, following the strategy from our previous work[24]. We retained sites with a minimum alignment mapping quality and base quality of 20, and downgraded the mapping quality for reads with excessive mismatches using a coefficient of 50. Sites with depth coverage significantly higher or lower than the average across the dataset (by a factor of two) were also masked, based on criteria from our prior research[24]. Additional filtering was applied using the MSMC mappability mask[22]. Low-quality positions in Altai Neanderthal and Denisovan genomes were excluded, as well as positions lacking ancestral allele information. This filtering left us with approximately 1.5 GB of regions with sufficient power for variant calling.

To reduce the power of selection acting on the genome, we also used a more stricter filtering strategy. In this strategy, we remove any positions which are within 20 kb of a gene defined in gencode version 42. This strategy left us with a region of ~370 mega base (mb) which is neutrally evolving.

We found that Site Frequency Spectrum (SFS) quickly becomes intractable with higher numbers of samples per population. For instance, using five samples from each modern population, along with one Neanderthal and one Denisovan, resulted in a dataset with 20,000 rows and 131,769 columns of simulated data. Where every row represents a different simulation run under varying parameter values, and each column corresponds to an element of a single SFS.

Since we included five samples from four human populations and one sample each from Neanderthal and Denisovan, the resulting SFS had six dimensions—one for each population. The length of each dimension depends on sample size:

Five-sample populations → Dimension length of 11 ($5 \times 2$ for diploid + 1 for no mutation count).

Single-sample populations → Dimension length of 3 (2 for diploid + 1 for no mutation count).

This six-dimensional array was then flattened into a one-dimensional array of elements per SFS:

$$11 \times 11 \times 11 \times 11 \times 3 \times 3 = 131,769$$

Although neural networks can handle such large datasets, parameter estimation is particularly computationally demanding. The parameter estimation needs multiple Sequential Monte Carlo (SMC) cycles to achieve precise estimates. A single cycle fully optimizes the neural network for the current set of simulations. However, the computation burden is immense as it has to optimise more than 33 million parameters ($131,769 \times 256$) only in the first hidden layer.

To address this, we used cross-population SFS (cSFS) as a more efficient alternative to classical SFS for training neural networks.

We generated two-dimensional SFS for each population pair. These were flattened and concatenated into a single-dimensional array. With our current six populations, we can have a total of 15 combinations of two populations:

- 6 modern human population pairs → Each with 121 elements ($11 \times 11$).
- 8 human-archaic pairs → Each with 33 elements ($11 \times 3$).
- 1 archaic-archaic pair → With 9 elements ($3 \times 3$).

The total cSFS length is:

$$(6 \times 121) + (8 \times 33) + 9 = 999$$

This approach reduced the amount of data to train significantly (e.g., from 131,769 to 999 columns more than 100 fold reduction), thereby improving the computational efficiency of neural network training.

We converted VCF files to SFS using the Run_VCF2SFS.py script, and then to cSFS using the Run_SFS_to_SFS2c.py script from ABC-

DLS[20]. We selected five individuals from each modern human population (African, European, East Asian, PNG), one Neanderthal, and one Denisovan sample. Multiple cSFS datasets were generated by varying the modern human samples while keeping the archaic samples constant. Ultimately, we obtained three cSFS datasets from 15 samples per modern human population and two archaics, which were concatenated row-wise. This multi-cSFS approach enhances parameter estimation by allowing median parameter values predicted from multiple cSFS datasets, thereby improving robustness in parameter estimation. Multiple cSFS is also useful for finding correct demographic models as the same top model produced by different cSFS suggests robustness of the model prediction.

**Simulation.** We used msprime v1.2.0[58] for simulating demographic models. These simulations generated cSFS datasets for training neural networks in ABC-DLS, and VCF files for input into Relate analysis. In all the models, we used the mutation rate of $1.25 \times 10^{-8}$ per base per generation, the recombination rate of $1 \times 10^{-8}$ per base per generation, and the generation time of 29 years per generation.

We tested five demographic models (Fig. 2) based on the framework of Gravel et al.[30], excluding migration rates between populations. All OOA populations experienced a strong bottleneck after diverging from African populations (Yoruba in this case). Neanderthals contributed genetic material to all OOA populations before the divergence of OOA populations[49], while Denisovans contributed only to PNG[7]. Both introgressed archaic populations diverged from the sequenced archaic populations long before these events[9,59]. OOA populations then experienced exponential growth after the separation of Europeans and East Asians[30].

**Model A.** The PNG population is a sister group to East Asian population[11,12].

**Model O.** The PNG population is an outgroup to European and East Asian populations[3,19].

**Model M.** A mixture of Models A and O, where the PNG population is a sister group to East Asians but also carries genetic contributions from an unknown outgroup population of European and East Asians.

**Model AX.** Similar to Model M, but the unknown contributing population diverged from Africans before the main OOA event, resembling the first OOA hypothesis[4].

**Model OX.** Similar to Model AX, but with the PNG population as an outgroup to European and East Asian populations.

Priors for these models (Supplementary Table 1) were chosen from our empirical results and previous studies. Initially, the priors don't need to have a narrow range. Our strategy of ABC works as long as the true parameter values fall within the limit of priors.

For instance, the modern effective population sizes were set as follows: European and East Asian populations ranged from 10,000 to 150,000, while African populations were set between 10,000 and 60,000, and PNG populations ranged from 10,000 to 50,000. This reflects the relatively modest population growth observed in the African and PNG populations in the Relate results, in contrast to the more pronounced population expansions in European and East Asian populations. The effective population sizes of the OOA populations prior to their exponential growth were set to much lower values, consistent with the well-documented bottleneck these populations experienced during the OOA event[6,30,60], which is also supported by our Relate results. Neanderthal and Denisovan populations were assumed to have smaller effective population sizes than modern humans, consistent with their known history of interbreeding[49,51]. The effective population size of the common ancestors of Neanderthals

and Denisovans was set broadly, ranging from 500 to 30,000, due to the limited information available about this population.

We assumed that the archaic introgression into the PNG population occurred most recently, between 10,000 and 100,000 years ago ($T_{DPM}$), based on the hypothesis that modern PNG population admixed with Denisovans after entering the Sunda and Sahul regions[52]. All other events in the model were assumed to occur consecutively before this timeframe, as detailed in Supplementary Table 7. For example, in Model A, the time interval of separation between PNG and East Asians is assumed to happen between 500 years and 50,000 years ago ($T_{EU\_AS}$), which corresponds to events ranging from 10,500 to 150,000 years ago in the simulation ($T_{DPM} + T_{EU\_AS}$). Whereas time intervals represent the parameter used in the simulation, events represent the timing of that event taking care of the past event that happened before that.

The level of introgression from archaic populations into modern humans was assumed to be between 1% and 5%, which is consistent with accepted estimates[49]. For the admixture from earlier OOA populations or outgroup Eurasian populations, we assumed a range of 1% to 99%, as analysis outside this range loses power to distinguish between models.

In certain scenarios, we incorporated migration rates between populations. Migration rates, denoted as m, represent the proportion of individuals moving from one population to another per generation, typically modelled as symmetric between populations. We applied these migration rates only to modern human populations (i.e., between Africans, Europeans, East Asians, and PNG) after their initial separation. The migration rates used in our models were $m_{ij} < 5 \times 10^{-5}$ and $m_{ij} < 5 \times 10^{-4}$, where i and j represent two different populations, and these values were used as priors.

**Minimal model of OOA.** To understand the effects of bottlenecks and exponential growth on RCCR curves, we simulated a minimal OOA model. African and OOA populations were set to have separated around 80 thousand years ago, each with an effective population size of 10,000 individuals. Europeans and PNG then diverged around 50 thousand years ago, with Europeans undergoing exponential growth (0.2% growth rate, leading to an effective population size of ~300,000 in modern times). The PNG population experienced various combinations of bottleneck sizes (500–10,000 individuals) and growth rates (0–0.2%) (Supplementary Table 6).

**ABC and neural network analysis.** We used ABC-DLS[20] for model comparison and parameter estimation.

**Model comparison.** We trained a neural network using 10,000 cSFS per model. For the simulations, we used 3 billion base pairs per sample, selecting five samples each from Yoruba, East Asian, European, and PNG populations, along with one sample from each archaic population (Altai-Neanderthal and Denisovan). The priors for the models are detailed in Supplementary Table 1. The neural network predictions were then incorporated into the Approximate Bayesian Computation (ABC) framework to assess the confidence of these predictions. We generated a total of three separate cSFS datasets and repeated the analysis 10 times. This resulted in 30 independent runs for each model comparison, ensuring robustness in our analysis.

**Parameter estimation.** For parameter estimation, we trained the neural network with 20,000 cSFS in a single SMC cycle. We began by running simulations with 100 MB (Mega Base) of sequence data per sample. As the simulation reached equilibrium, we progressively increased the sequence length to 500 MB, 1.5 GB, and finally 3 GB per sample, optimising the parameter estimates at each stage. We gradually increased the simulated region in the msprime during the SMC cycles for two key reasons. First, it reduced simulation time in the initial stages when the parameter space was vast, making it inefficient

to simulate the full 3 GB region upfront. Second, this approach initially lowers precision, which helps improve the accuracy of predicted parameter ranges by reducing the risk of getting trapped in local minima of parameter space.

**Neural network architecture.** We implemented our neural network architecture using TensorFlow with a Keras backend[61]. The model consisted of four fully connected hidden layers with 256, 128, 64, and 32 neurons, respectively. Each layer employed the Rectified Linear Unit (ReLU) activation function. We applied a 1% dropout rate between each hidden layer to lower the overfitting. Furthermore, we introduced random Gaussian noise (with standard deviation of 0.05) to the input data to further reduce the overfitting. Gaussian noise can introduce negative numbers in our training dataset which was rectified using the ReLU layer before the data goes through the hidden layers.

The architecture remained the same for both classification and parameter estimation, except for the output layer. For classification, the output layer contained five neurons (corresponding to model categories) with a softmax activation function. For parameter estimation, the output layer had 24 neurons (matching the number of parameters) with a linear activation function. We used categorical cross-entropy as the loss function for classification and the logarithm of the hyperbolic cosine (log-cosh) for parameter estimation.

For optimization, we employed the Adam optimizer for classification and Nadam for parameter estimation. The validation dataset contained 10,000 samples in both cases, while training datasets differed: 40,000 samples for classification and 10,000 for parameter estimation.

The learning rate was initially set to 0.001 and halved if no improvement was observed in validation loss over 10 epochs. Early stopping was applied, terminating training if validation loss did not improve for 100 consecutive epochs. We retained only the model with the lowest validation loss using model checkpointing.

To further reduce overfitting, we shuffled data at every epoch to randomize sample order. All training used the default batch size of 32.

**ChatGPT**
ChatGPT was used exclusively to enhance sentence clarity and correct grammar. We utilised the "Write For Me" customised version of Generative Pre-trained Transformer (GPT) based on the ChatGPT-4 framework, providing prompts such as "improve the following text:". Each sentence generated by ChatGPT was manually reviewed and only incorporated if it improved the readability compared to the original version. ChatGPT was not involved in any aspect of the scientific analysis.

**Reporting summary**
Further information on research design is available in the Nature Portfolio Reporting Summary linked to this article.

## Data availability
No data was generated for this manuscript. We used PNG data available in EGA under accession numbers EGAD00001010142, EGAD00001010143, and EGAD50000000050. The 1000 genome data was downloaded from the specific website (https://www.internationalgenome.org/data-portal/data-collection/30x-grch38). The Andamanese data set is downloaded from ENA with accession number PRJEB11455. Neranderthal genome was downloaded from http://cdna.eva.mpg.de/neandertal/altai/AltaiNeandertal/VCF/ and Denisova genome was downloaded from http://cdna.eva.mpg.de/denisova/VCF/. Ancestral fasta file is downloaded from ensemble website (https://ftp.ensembl.org/pub/release-105/fasta/ancestral_alleles/homo_sapiens_ancestor_GRCh38.tar.gz). The GRCh38 genome reference was built into the GATK workflow (https://console.cloud.google.com/storage/browser/ genomics-public-data/resources/

broad/hg38/v0/). The source data for the figures and empirical cSFS generated in this manuscript can be found in figshare repository under 29222954.

## Code availability
All the custom codes to generate this particular manuscript can be found in https://github.com/mayukhmondal/png_xOOA [62]. Source code for ABC-DLS can be found at https://github.com/mayukhmondal/ABC-DLS [63].

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

## Acknowledgements

This research was supported by the European Union through the European Regional Development Fund (Project No. 2014-2020.4.01.16-0030) to M.A. and through Horizon 2020 research and innovation programme under grant no 810645 and the European Regional Development Fund project no. MOBEC008 to A.E., M.A., M.Metspalu and M.Mondal. A.K.P. was supported by the Estonian Research Council grant PUT (PUTJD1186). M.Metspalu was supported by the Estonian Research Council grant PUT (PRG1899), the European Union's Horizon Europe research and innovation programme under grant agreement No 101060011 (views and opinions expressed are however those of the author(s) only and do not necessarily reflect those of the European Union or European Research Executive Agency. Neither the European Union

nor the granting authority can be held responsible for them and the Estonian Research Council grant TK (TK214). F.-X.R. and N. B. were supported by the French Nationale Research Agency grant ANR-20-CE12-0003-01. Data analyses were carried out in part in the High-Performance Computing Center of the University of Tartu and supported by Tartu University grant no. PLTGI22904 to A.E.

## Author contributions

M.Mo., M.Me. and A.E. designed the study. M.Mo., M.A. and A.K.P. performed data analysis. M.Mo., M.A., A.K.P., N.B., F.-X.R., M.Me. and A.E. helped in writing the manuscript.

## Competing interests

The authors declare no competing interests.
