## [Peer Review file · Nature Communications]

Resolving out of Africa event for Papua New Guinean population using neural network

Corresponding Author: Dr Mayukh Mondal

Version 0:

Reviewer comments:

Reviewer #1

(Remarks to the Author)

The authors utilized previously published data to investigate the demographic relationships between Papua New Guinea (PNG) and other human populations. Their analysis included samples from the Yoruba, British (from England and Scotland), Han Chinese South, PNG highlander and lowlander groups, and the Andamanese population, with 40 random samples per population, except for the Andamanese, which had only 10 samples per group.

Using Relate, the authors estimated divergence times between PNG and African populations, identifying a split occurring approximately 100,000 years ago. This finding aligns with previous studies and supports the hypothesis of an early Out of Africa migration. They argue that these results are unlikely to be artifacts of phasing errors, stating that “the likelihood of systematic phasing errors causing this shift is minimal.” A similar pattern was observed when applying Relate to the Andamanese population.

To further interpret these results, the authors tested five demographic models. These included models where (1) PNG and East Asians are sister groups, (2) PNG is an outgroup to other Out of Africa populations, (3) PNG represents an admixture between an outgroup and Out of Africa populations, and (4) PNG results from an early human population.

They then trained artificial neural networks (ANNs) using the site frequency spectrum between population pairs as input, predicting the model that generated the simulations. This was complemented by approximate Bayesian computation (ABC). The model with the strongest support positioned PNG and East Asians as sister groups. Parameter estimates aligned with previous studies, except for evidence of a pronounced bottleneck in the PNG population and significantly lower growth rates compared to other Out of Africa populations. Simulated datasets using Relate showed similar patterns to those observed in real data.

Additional analyses revealed that the RCCR curves were affected by the severity of bottlenecks and population growth rates.

Two main findings emerge from this study. First, on the theoretical side of population genomics, the authors highlight that RCCR interpretation can be misleading under conditions of severe bottlenecks. Second, they confirm that PNG and East Asians are sister groups, countering the hypothesis that PNG resulted from an early Out of Africa event. However, they caution that the use of cSFS can misclassify demographic models when migration rates are relatively high.

I have some issues regarding the details on how the study has been conducted.

- 1) Fitted Demographic Model: To what extent does the fitted demographic model replicate the summary statistics from the observed data in an unseen sample of individuals? I would suggest to generate simulations, compute the cSFS and estimate the distance between the cSFS and the cSFS in the observed data. This distance matrix could then be projected in a lower dimensional space using for example multidimensional scaling, and check the position of the observed data in the cloud of points from the simulated datasets.
- 2) Chromosome Selection: In line 278, why is only chromosome one considered, despite the authors stating that all chromosomes were used and that results are consistent across chromosomes?
- 3) Archaic Genomes: In the Data section of the Materials and Methods, it would be helpful to introduce the archaic genomes used in the study.
- 4) Redundant Wording: In line 287, the word “from” appears twice and should be removed.
- 5) DL Architecture Explanation: In line 417, even though it may be explained in the original article, please provide a brief explanation of the deep learning architecture used.
- 6) Simulation Parameters: What mutation rate, recombination rate, and generation time were used in the simulations?
- 7) Selection in Simulations: From lines 419 and 429, it seems that 3Gb of nucleotides are simulated. How were background

selection and positive selection accounted for in these simulations?
8) Figure 2: The Npad parameter appears to be misplaced in Figure 2.

(Remarks on code availability)

This code has been already proposed and used in another paper, which is properly referred in the manuscript.

Reviewer #2

(Remarks to the Author)

In this manuscript, the authors applied a tool called ABC-DLS, which utilizes artificial neural networks, to revisit the demographic history of the Papua New Guinean (PNG) population. A long-standing debate in human evolution is whether Papuans represent an outgroup to Europeans and Asians or form a sister lineage to Asians. This study employed machine learning to address this question, highlighting its potential as a novel approach in population genetics. However, I feel the manuscript lacks sufficient details on their methods and does not thoroughly examine various confounding factors. As a result, the conclusion that Papuans are a sister group to Asians is, in my view, not entirely convincing.

Major concerns:

1. Artificial neural networks (ANNs) are central to this study, as highlighted by the title's emphasis on "using neural network." However, the authors did not comprehensively document how they trained the neural networks. They merely stated that a neural network was trained (Line 419), without providing sufficient details. Given that the performance of ANNs is influenced by numerous factors, the authors should clearly report the training process, including the architecture of the neural network (e.g., the number of layers, the number of neurons per layer, and the activation functions), the choice of loss function and optimization algorithm, the hyperparameters used (such as learning rate, batch size, and number of epochs), the method for evaluating performance during training (e.g., validation dataset and metrics used), and any measures taken to address overfitting (e.g., regularization techniques, dropout, or early stopping). Providing these details would greatly enhance the transparency and reproducibility of the study.
2. In Lines 324–334, the authors discuss the challenges of using SFS. However, the meaning of "columns" and "rows" (Lines 326–327) is unclear, as is how the numbers "131,769" (Line 326) and "999" (Line 333) were determined. I recommend revising this section to clarify these points for better reader comprehension. Additionally, the term "cycle" is used in Line 329 and again in Line 426 in the phrase, "For parameter estimation, we trained the neural network with 20,000 cSFS in a single cycle." Does "cycle" refer to an epoch or batch size? I suggest revising this term to align with standard machine learning terminology to ensure clarity and avoid potential ambiguity.
3. It appears that the authors created only three different cSFS datasets (Lines 339–341). Given that populations in the 1000 Genomes Project typically include around 100 individuals, and the authors sampled only five individuals from each population, it is unclear why only three cSFS datasets were tested. Were these datasets sufficient to capture the variability and ensure robustness in the analysis? I recommend the authors provide a justification for this choice and, if possible, explore whether additional cSFS datasets would affect the robustness of their conclusions.
4. The authors stated that they used msprime for simulation (Line 347). However, in Lines 427–429, they describe progressively increasing the sequence length after the simulation reached equilibrium. To my knowledge, msprime does not require this approach to reach equilibrium, as this is typically associated with SLiM simulations. Additionally, msprime is capable of simulating long sequences directly. It also appears that the authors did not account for varying mutation rates and recombination rates along the genome. Could the authors elaborate on why they chose to progressively increase the sequence length in msprime? Understanding the rationale behind this decision would greatly enhance the clarity of the methods section. Publishing their simulation code would also be valuable for understanding their simulation approach and ensuring reproducibility.
5. The authors applied ABC-DLS for analysis (Line 417), but no description is provided on how this method performs model comparison and parameter estimation. Currently, readers must refer to their previous publication for details. Including a brief explanation would allow readers to understand the method without relying on external sources.
6. The authors' primary support for their argument is that the RCCR curves from simulated data closely resemble those from real data. However, these curves are presented in separate figures, making direct comparison challenging. I recommend combining the RCCR curves from real and simulated data into a single figure to facilitate easier and clearer comparison. Additionally, while RCCR is a valuable tool for inferring demographic history, it may not fully capture the complexity of population dynamics. To strengthen their argument, the authors could also compare other summaries, such as the site frequency spectrum and linkage disequilibrium, to ensure that their model aligns with multiple aspects of the data.
7. The authors observed that high migration rates can make their approach fail to distinguish whether the PNG population is an outgroup of Europeans and Asians or a sister group to Asians (Lines 131–138). A previous study (https://popsim-consortium.github.io/stdpopsim-docs/stable/catalog.html#sec_catalog_homsap_models_papuansoutofafrica_10j19) suggests a higher migration rate (10^{-3}) between East Asians and Papuans compared to the rates used in this study (10^{-4} and 10^{-5}). However, the authors chose to model without migration (Lines 233–234), which I believe is unrealistic, as it contradicts archaeological evidence. Both biological and historical perspectives are essential for studying human evolutionary history. Incorporating archaeological and genetic evidence into the discussion would enhance the study's historical context and biological relevance.
8. Finally, the authors did not consider the potential effects of natural selection. They suggest that the PNG population experienced a significant bottleneck and slower population growth, which could explain the observed low genetic diversity. However, strong background selection can also lead to a similar reduction in genetic diversity. The authors only explored demographic history under the assumption of neutral evolution and did not discuss how natural selection might influence their analysis. I recommend addressing this limitation and considering the potential impact of natural selection on their findings.

Minor concerns:

1. Asian populations encompass a wide range of groups with significant population divergence. Since the authors specifically used Han Chinese in their study, I recommend replacing "Asians" with "East Asians" for greater specificity and accuracy.
2. The authors did not specify the mutation rate or recombination rate used in their simulation. Providing these values is essential for ensuring reproducibility and should be included in the manuscript.
3. In Lines 287–288, the phrase "Thus we downloaded GVCF files coming from from the high coverage 1000 genome dataset" contains a duplicated "from." I recommend removing one to correct this error.
4. The authors tested five different demographic models (Lines 358–371); however, only a text description is provided. Including visualizations of these models would greatly enhance the readers' understanding.

(Remarks on code availability)

Although the authors claim that the code is available at <https://github.com/mayukhmondal/ABC-DLS>, this repository only contains the source code for ABC-DLS itself. The code used for simulation and analysis is not included. Given the complexity of machine learning approaches and their sensitivity to various factors, I recommend that the authors publish the full set of code used for their study, particularly the simulation code. This will enable readers to better evaluate and replicate their study. Additionally, implementing the workflow using modern workflow managers such as NextFlow or Snakemake would further enhance reproducibility.

Reviewer #3

(Remarks to the Author)

Based on the relative cross coalescent rate (RCCR) between Papuan New Guineans (PNG) and other populations, it was suggested in Pagani et al 2016 (Nature 538) that PNG has some of their ancestry from a population that migrated out of Africa earlier than the out of Africa event that gave rise to all other non-African populations. Here, the authors (some of which are co-authors of the Pagani et al paper) recreate the RCCR patterns between PNG and other populations using more data and convincingly argue that this is not an artifact showing that this is a robust and real signal. However, using the latest ABC-techniques they find that the most likely demographic scenario does not include an earlier out of Africa event. Moreover, the most likely simulated demographic history includes a bottleneck with slower recovery compared to other non-African population and does, indeed, generate the same RCCR pattern as is observed for the real data. Although they cannot rule out a small contribution from an earlier out of Africa event to PNG, the authors show that there is no longer a need to invoke such an event to explain the data. Importantly, they show that the timing of the bottleneck in PNG is very poorly correlated with the time of the effect in the RCCR-signal.

I feel honored to be a reviewer of this paper and feel that this is truly a strong paper (especially considering that some of the authors are co-authors on the Pagani et al paper) that is mitigating the sometimes non-conservative ways of science. I have no major revisions but encourage the authors to follow up on the implications for the interpretation of RCCR curves. The results of this article mirrors the analysis in Mazet et al 2016 (Heredity 116:362-371) where they investigate the effect of population structure on PSMC analyses and found a similar non-correspondence in time between the demographic event and the visible effect on the PSMC curve.

Some minor things:

Line 135 Does Bayes factor really translate into significance levels?

Line 174-175 Do you mean "500 individuals"? I don't understand what is meant by "500 samples" in this context.

Line 374 What is meant by "The priors do not have to be exact"

(Remarks on code availability)

Version 1:

Reviewer comments:

Reviewer #1

(Remarks to the Author)

The authors have adequately addressed all of my comments.

(Remarks on code availability)

Implementing the whole pipeline would require at least several months of running time in a cluster.

Reviewer #2

(Remarks to the Author)

(Remarks on code availability)

REVIEWER COMMENTS

Reviewer #1 (Remarks to the Author):

The authors utilized previously published data to investigate the demographic relationships between Papua New Guinea (PNG) and other human populations. Their analysis included samples from the Yoruba, British (from England and Scotland), Han Chinese South, PNG highlander and lowlander groups, and the Andamanese population, with 40 random samples per population, except for the Andamanese, which had only 10 samples per group.

Using Relate, the authors estimated divergence times between PNG and African populations, identifying a split occurring approximately 100,000 years ago. This finding aligns with previous studies and supports the hypothesis of an early Out of Africa migration. They argue that these results are unlikely to be artifacts of phasing errors, stating that “the likelihood of systematic phasing errors causing this shift is minimal.” A similar pattern was observed when applying Relate to the Andamanese population.

To further interpret these results, the authors tested five demographic models. These included models where (1) PNG and East Asians are sister groups, (2) PNG is an outgroup to other Out of Africa populations, (3) PNG represents an admixture between an outgroup and Out of Africa populations, and (4) PNG results from an early human population.

They then trained artificial neural networks (ANNs) using the site frequency spectrum between population pairs as input, predicting the model that generated the simulations. This was complemented by approximate Bayesian computation (ABC). The model with the strongest support positioned PNG and East Asians as sister groups. Parameter estimates aligned with previous studies, except for evidence of a pronounced bottleneck in the PNG population and significantly lower growth rates compared to other Out of Africa populations. Simulated datasets using Relate showed similar patterns to those observed in real data.

Additional analyses revealed that the RCCR curves were affected by the severity of bottlenecks and population growth rates.

Two main findings emerge from this study. First, on the theoretical side of population genomics, the authors highlight that RCCR interpretation can be misleading under conditions of severe bottlenecks. Second, they confirm that PNG and East Asians are sister groups, countering the

hypothesis that PNG resulted from an early Out of Africa event. However, they caution that the use of cSFS can misclassify demographic models when migration rates are relatively high.

I have some issues regarding the details on how the study has been conducted.

1) Fitted Demographic Model: To what extent does the fitted demographic model replicate the summary statistics from the observed data in an unseen sample of individuals? I would suggest to generate simulations, compute the cSFS and estimate the distance between the cSFS and the cSFS in the observed data. This distance matrix could then be projected in a lower dimensional space using for example multidimensional scaling, and check the position of the observed data in the cloud of points from the simulated datasets.

Reply: We thank the reviewer for this valuable suggestion. Model selection alone is insufficient if the demographic model's output does not align well with the empirical data. To address this, we used the goodness-of-fit test implemented in the ABC package. Our results show that the best-fit model (Model A) has a p-value > 0.05 , indicating a good fit with the observed data. We have included the goodness-of-fit p-value in Supplementary Table 2 and in all model selection tables throughout the manuscript. The relevant line in the manuscript has also been revised accordingly.

Page 5 line 123: "with a high p value (>0.05) for goodness of fit analysis"

2) Chromosome Selection: In line 278, why is only chromosome one considered, despite the authors stating that all chromosomes were used and that results are consistent across chromosomes?

Reply: Whole-genome analyses are generally more robust in RELATE due to the larger data available for the programme. However, running RELATE on whole-genome sequences for 160 individuals (40 individuals from each of four populations), with 10 independent resampling runs, is computationally intensive. One of the main goals of this study was to reproduce the shift in the RCCR curve. Since we observed no substantial difference between chromosome 1 and the whole genome regarding this shift, we did not conduct whole-genome runs for every analysis. Chromosome 1, given its length and with 160 samples, provided results comparable in robustness to whole-genome data. In contrast, smaller chromosomes alone (for example

chromosome 22) or smaller sample sizes (2 or 9) were insufficiently informative and thus not used. We have revised the sentence in the manuscript to clarify this point.

Page 11 Line 314: “chromosome 1 might not be enough to get robust results.”

3) *Archaic Genomes: In the Data section of the Materials and Methods, it would be helpful to introduce the archaic genomes used in the study.*

Reply: We added few line as the reviewer requested:

Page 12 Line 350: “Both Altai Neanderthal and Denisovan are two ancient ..”

4) *Redundant Wording: In line 287, the word “from” appears twice and should be removed.*

Reply: Thank you very much for noticing it. We removed it.

5) *DL Architecture Explanation: In line 417, even though it may be explained in the original article, please provide a brief explanation of the deep learning architecture used.*

Reply: This is indeed very crucial. Thanks for pointing it out. We added a paragraph where we explained the underlying neural network architecture is used.

Page 18 Line 532: “Neural Network Architecture..”

6) *Simulation Parameters: What mutation rate, recombination rate, and generation time were used in the simulations?*

Reply: We used a mutation rate of 1.25×10^{-8} per base pair per generation, recombination rate of 1×10^{-8} per base per generation and generation of 29 years were used. We now mentioned that in the manuscript.

Page 15 Line 439: “In all the models, we used the mutation rate of 1.25×10^{-8} per base per generation, the recombination rate of 1×10^{-8} per base per generation and the generation time of 29 years per generation.”

7) *Selection in Simulations: From lines 419 and 429, it seems that 3Gb of nucleotides are simulated. How were background selection and positive selection accounted for in these simulations?*

Reply: All our analyses are based on neutral simulations. We acknowledge that background selection and positive selection in empirical data could introduce bias. To assess the impact, we applied an additional mask to the empirical data—on top of the existing mask, we excluded all positions within ± 10 kb of Ensembl genes. The results remained consistent, suggesting that the level of background and positive selection present in the human genome does not significantly

bias our findings. We have included these results in Supplementary Table 3 and added a corresponding line to the manuscript.

Page 5 Line 123: "This result held even after excluding..."

8) *Figure 2: The Npad parameter appears to be misplaced in Figure 2.*

Reply: Thanks. We have updated our Figure 2.

Reviewer #1 (Remarks on code availability):

This code has been already proposed and used in another paper, which is properly referred in the manuscript.

Reviewer #2 (Remarks to the Author):

In this manuscript, the authors applied a tool called ABC-DLS, which utilizes artificial neural networks, to revisit the demographic history of the Papua New Guinean (PNG) population. A long-standing debate in human evolution is whether Papuans represent an outgroup to Europeans and Asians or form a sister lineage to Asians. This study employed machine learning to address this question, highlighting its potential as a novel approach in population genetics. However, I feel the manuscript lacks sufficient details on their methods and does not thoroughly examine various confounding factors. As a result, the conclusion that Papuans are a sister group to Asians is, in my view, not entirely convincing.

Major concerns:

1. *Artificial neural networks (ANNs) are central to this study, as highlighted by the title's emphasis on "using neural network." However, the authors did not comprehensively document how they trained the neural networks. They merely stated that a neural network was trained (Line 419), without providing sufficient details. Given that the performance of ANNs is influenced by numerous factors, the authors should clearly report the training process, including the architecture of the neural network (e.g., the number of layers, the number of neurons per layer,*

and the activation functions), the choice of loss function and optimization algorithm, the hyperparameters used (such as learning rate, batch size, and number of epochs), the method for evaluating performance during training (e.g., validation dataset and metrics used), and any measures taken to address overfitting (e.g., regularization techniques, dropout, or early stopping). Providing these details would greatly enhance the transparency and reproducibility of the study.

Reply: We thank the reviewer for pointing out this crucial point. We have updated our manuscript to include the neural network architecture in the method section in the manuscript:

Page 18 Line 532: “Neural Network Architecture”

2. In Lines 324–334, the authors discuss the challenges of using SFS. However, the meaning of “columns” and “rows” (Lines 326–327) is unclear, as is how the numbers “131,769” (Line 326) and “999” (Line 333) were determined. I recommend revising this section to clarify these points for better reader comprehension. Additionally, the term “cycle” is used in Line 329 and again in Line 426 in the phrase, “For parameter estimation, we trained the neural network with 20,000 cSFS in a single cycle.” Does “cycle” refer to an epoch or batch size? I suggest revising this term to align with standard machine learning terminology to ensure clarity and avoid potential ambiguity.

Reply: We have updated the paragraph to make it easier to understand how the numbers were calculated. Hope this clarifies the point more clearly.

Page 14 Line 395: “Since we included five samples...”

The cycle here is SMC (Sequential Monte Carlo) cycles. In one cycle of SMC, we totally optimised the neural network for the current set of simulated data (cSFS). We agree it was not clear which cycle here we referenced, especially the reviewer correctly pointing out that there may be several different types of cycles. We have updated our manuscript.

Page 17 Line 522: “in a single SMC cycle.”

3. It appears that the authors created only three different cSFS datasets (Lines 339–341). Given that populations in the 1000 Genomes Project typically include around 100 individuals, and the authors sampled only five individuals from each population, it is unclear why only three cSFS datasets were tested. Were these datasets sufficient to capture the variability and ensure robustness in the analysis? I recommend the authors provide a justification for this choice and, if

possible, explore whether additional cSFS datasets would affect the robustness of their conclusions.

Reply: We used three different cSFS datasets, rather than relying on a single one, to test the robustness of our analysis. Each cSFS yields its own model probability due to the way ABC is calculated. In principle, a single cSFS is sufficient if there is enough power to distinguish between models, which aligns with our previous experience.

To further validate our results, we ran 10 independent neural network analyses for each cSFS—30 runs in total—and obtained consistent results across all instances. Overall, our method proved highly robust; the results remained stable across different filtering strategies and even datasets demonstrated in our previous study (Montinaro 2021, AJHG). The only exception occurred when we introduced higher migration rates, where different models emerged as top candidates. This was expected, as our results showed that in such cases, our model inference loses power.

We mentioned this phenomenon in the results:

Page 7 Line 198: “This result was reproduced across..”

4. The authors stated that they used msprime for simulation (Line 347). However, in Lines 427–429, they describe progressively increasing the sequence length after the simulation reached equilibrium. To my knowledge, msprime does not require this approach to reach equilibrium, as this is typically associated with SLiM simulations. Additionally, msprime is capable of simulating long sequences directly. It also appears that the authors did not account for varying mutation rates and recombination rates along the genome. Could the authors elaborate on why they chose to progressively increase the sequence length in msprime? Understanding the rationale behind this decision would greatly enhance the clarity of the methods section. Publishing their simulation code would also be valuable for understanding their simulation approach and ensuring reproducibility.

Reply: We initially used shorter sequence lengths in msprime to accelerate early SMC cycles. At the start, the parameter space is large, and simulating across the full space at high precision would be time-consuming. By simulating smaller regions early on, we reduced runtime with minimal precision, which is acceptable at this exploratory stage.

As the analysis progressed, we gradually increased the sequence length in msprime. This increased simulation time but also improved precision, allowing for a more refined search. This stepwise approach not only reduced total runtime but also minimized the risk of SMC getting

stuck in local minima due to poor predictions in early cycles. We apologize for the lack of clarity in the initial draft and have revised the manuscript to better explain this method.

Page 18 Line 526: “We gradually increased the simulated..”

We also publicly share the whole code that was used to produce the results. Hope this helps as well:

https://github.com/mayukhmondal/png_xOOA

5. The authors applied ABC-DLS for analysis (Line 417), but no description is provided on how this method performs model comparison and parameter estimation. Currently, readers must refer to their previous publication for details. Including a brief explanation would allow readers to understand the method without relying on external sources.

Reply: We added a paragraph to explain the ABC-DLS method in the methods section.

Page 11 Line 322: “Approximate Bayesian Computation with Deep Learning ...”

6. The authors’ primary support for their argument is that the RCCR curves from simulated data closely resemble those from real data. However, these curves are presented in separate figures, making direct comparison challenging. I recommend combining the RCCR curves from real and simulated data into a single figure to facilitate easier and clearer comparison. Additionally, while RCCR is a valuable tool for inferring demographic history, it may not fully capture the complexity of population dynamics. To strengthen their argument, the authors could also compare other summaries, such as the site frequency spectrum and linkage disequilibrium, to ensure that their model aligns with multiple aspects of the data.

Reply: This is a valuable suggestion. However, we believe including too many lines in the plot could confuse the readers and detract from the key result—the shift in the RCCR curve—which we aim to highlight. Therefore, we placed the empirical and simulated data comparisons in Supplementary Figures 13, 14 and 15. We also included a comparison of the derived allele frequency spectrum in Supplementary Figure 16.

In our ABC-DLS analysis, we used the site frequency spectrum (SFS), which is known to be unaffected by recombination rate. This choice of summary statistics ensures that model comparison or parameter inference is independent of linkage disequilibrium effects, so we did not include LD-based comparison plots. We have clarified this in the manuscript:

Page 8 Line 203: “Model A, with the best-fit parameter..”

7. The authors observed that high migration rates can make their approach fail to distinguish whether the PNG population is an outgroup of Europeans and Asians or a sister group to Asians (Lines 131–138). A previous study (https://popsim-consortium.github.io/stdpopsim-docs/stable/catalog.html#sec_catalog_homsap_models_papuansoutofafrica_10j19) suggests a higher migration rate (10^{-3}) between East Asians and Papuans compared to the rates used in this study (10^{-4} and 10^{-5}). However, the authors chose to model without migration (Lines 233–234), which I believe is unrealistic, as it contradicts archaeological evidence. Both biological and historical perspectives are essential for studying human evolutionary history. Incorporating archaeological and genetic evidence into the discussion would enhance the study's historical context and biological relevance.

Reply: The page was recently updated, and it now reports a much lower migration rate between East Asians and Papuans— 5.72×10^{-5} —which is only slightly higher than the rate we used in Supplementary Table 3. The main difference between our model and theirs lies in their use of high migration rates between ancestral populations (e.g., Ghost-Africa, Ghost-Europe, Eurasia-Papuan, Eurasia-Ghost), which we did not include.

We deliberately avoided modeling high migration between ancestral populations because we lack sufficient knowledge about them, and there is no direct method to estimate these rates. While MSMC-IM can estimate migration rates, it applies to modern populations, not to ancestral or ghost populations. Without a reliable estimation method or even lacking an ancient genome, including ancestral migration rates risks overfitting the model parameters, especially if the underlying model is incorrect. We have added a few paragraphs to discuss this important point in the manuscript.

Page 9 Line 256: “There are two competing hypotheses..”

8. Finally, the authors did not consider the potential effects of natural selection. They suggest that the PNG population experienced a significant bottleneck and slower population growth, which could explain the observed low genetic diversity. However, strong background selection can also lead to a similar reduction in genetic diversity. The authors only explored demographic history under the assumption of neutral evolution and did not discuss how natural selection might influence their analysis. I recommend addressing this limitation and considering the potential impact of natural selection on their findings.

Reply: We agree that regions under selection could, in principle, affect our results, as our simulations are based solely on neutral regions, while the empirical data includes both positively and negatively selected regions. This could introduce bias in our inferences. However, in

practice, we found no such effect. To minimize the impact of selection, we used a stricter mask that excluded all genes with 10 kb flanking region (as also noted by another reviewer; see above). Even with this more conservative approach, our results remained unchanged (Supplementary Table 3), likely because the proportion of selected regions is small relative to the overall neutrally evolving genome.

Page 5 Line 123: “This result held even after excluding...”

Minor concerns:

1. *Asian populations encompass a wide range of groups with significant population divergence. Since the authors specifically used Han Chinese in their study, I recommend replacing “Asians” with “East Asians” for greater specificity and accuracy.*

Reply: We agreed with the reviewer and updated our manuscript.

2. *The authors did not specify the mutation rate or recombination rate used in their simulation. Providing these values is essential for ensuring reproducibility and should be included in the manuscript.*

Reply: We have updated the manuscript mentioning the rates:

Page 15 Line 439: “In all the models, we used mutation rate of 1.25×10^{-8} per base pair per generation, recombination rate of 1×10^{-8} per base per generation and generation time of 29 years per generation.”

3. *In Lines 287–288, the phrase “Thus we downloaded GVCF files coming from from the high coverage 1000 genome dataset” contains a duplicated “from.” I recommend removing one to correct this error.*

Reply: Thanks for finding this error. We have removed it.

4. *The authors tested five different demographic models (Lines 358–371); however, only a text description is provided. Including visualizations of these models would greatly enhance the readers’ understanding.*

Reply: We agree. We moved the schematic from the supplementary to the main Figure 2.

Reviewer #2 (Remarks on code availability):

Although the authors claim that the code is available at <https://github.com/mayukhmondal/ABC-DLS>, this repository only contains the source code for ABC-DLS itself. The code used for simulation and analysis is not included. Given the complexity of machine learning approaches and their sensitivity to various factors, I recommend that the authors publish the full set of code used for their study, particularly the simulation code. This will enable readers to better evaluate and replicate their study. Additionally, implementing the workflow using modern workflow managers such as NextFlow or Snakemake would further enhance reproducibility.

Reply: We have now shared all the code necessary to reproduce our results. Hope this helps now.

https://github.com/mayukhmondal/png_xOOA

Reviewer #3 (Remarks to the Author):

Based on the relative cross coalescent rate (RCCR) between Papuan New Guineans (PNG) and other populations, it was suggested in Pagani et al 2016 (Nature 538) that PNG has some of their ancestry from a population that migrated out of Africa earlier than the out of Africa event that gave rise to all other non-African populations. Here, the authors (some of which are co-authors of the Pagani et al paper) recreate the RCCR patterns between PNG and other populations using more data and convincingly argue that this is not an artifact showing that this is a robust and real signal. However, using the latest ABC-techniques they find that the most likely demographic scenario does not include an earlier out of Africa event. Moreover, the most likely simulated demographic history includes a bottleneck with slower recovery compared to other non-African population and does, indeed, generate the same RCCR pattern as is observed for the real data. Although they cannot rule out a small contribution from an earlier out of Africa event to PNG, the authors show that there is no longer a need to invoke such an event to explain the data.

Reply: Thank you very much. This is indeed the best way to summarise the results. We really liked the last sentence and adapted it in our abstract to reflect it.

Importantly, they show that the timing of the bottleneck in PNG is very poorly correlated with the time of the effect in the RCCR-signal.

I feel honored to be a reviewer of this paper and feel that this is truly a strong paper (especially considering that some of the authors are co-authors on the Pagani et al paper) that is mitigating the sometimes non-conservative ways of science. I have no major revisions but encourage the authors to follow up on the implications for the interpretation of RCCR curves. The results of this article mirrors the analysis in Mazet et al 2016 (Heredity 116:362-371) where they investigate the effect of population structure on PSMC analyses and found a similar non-correspondence in time between the demographic event and the visible effect on the PSMC curve.

Some minor things:

Line 135 Does Bayes factor really translate into significance levels?

Reply: Thanks for this comment. Indeed Bayes factor does not translate to significance level it only shows that model A is more likely than other models or not. We updated the line:

Page 5 Line 142: "the evidence for Model O "

Line 174-175 Do you mean "500 individuals"? I don't understand what is meant by "500 samples" in this context.

Reply: This is indeed correct. We have updated the line.

Line 374 What is meant by "The priors do not have to be exact".

Reply: We meant to say that we do not need to start with a narrow prior (matching exactly with the parameter values for empirical data). We have updated the line.

Page 16 Line 465: "Initially, the priors don't need to have a narrow range."